# The Atmospheric Vertical Detection of Large Area Regions Based on Interference Signal Denoising of Weighted Adaptive Kalman Filter

**DOI:** 10.3390/s22228724

**Published:** 2022-11-11

**Authors:** Qiying Shen, Yongsheng Liu, Ren Chen, Zhijing Xu, Yuan Zhang, Yaxuan Chen, Jingyu Huang

**Affiliations:** 1School of Mathematics and Physics, Shanghai University of Electric Power, Shanghai 201306, China; 2Shanghai Institute of Technical Physics, Chinese Academy of Sciences, Shanghai 200083, China; 3Institute of Solar Energy, Shanghai University of Electric Power, Shanghai 200090, China; 4Key Laboratory of Infrared System Detection and Imaging Technology, Chinese Academy of Sciences, Shanghai 200083, China; 5College of Information Engineering, Shanghai Maritime University, Shanghai 201306, China; 6School of Optical-Electrical and Computer Engineering, University of Shanghai for Science and Technology, Shanghai 200082, China; 7School of Physics and Electronic Sciences, East China Normal University, Shanghai 200241, China

**Keywords:** focal plane array, noise, Kalman filter, interference signal, interferogram

## Abstract

In comparison with traditional space infrared spectroscopy technology, the interference signals of a large focal plane array (FPA) can be used to obtain spectra over a larger area range and rapidly achieve large-scale coverage of hyperspectral remote sensing. However, the low signal-to-noise ratio of the interference signals limits the application of spectral data, especially when atmospheric detection occurs in the long-wavelength infrared (LWIR) band. In this paper, we construct an LWIR hyperspectral system of a Fourier transform spectrometer composed of a HgCdTe photovoltaic IR FPA and a Michelson interferometer. The LWIR interference signals are obtained by a high-frequency oversampling technique. We use the Kalman filter (KF) and its improved weighted adaptive Kalman filter (WAKF) to reduce the noise of multiple measured data of each pixel. The effect of overshoot and ringing artifacts on the objective signals is reduced by the WAKF. The applicability is studied by the interference signals from the different sampling frequencies and different pixels. The effectiveness is also verified by comparing the spectra of denoised interferograms with the reference spectrum. The experimental results show that the WAKF algorithm has excellent noise suppression, and the standard deviation of the interferogram can be reduced by 39.50% compared with that of KF. The WAKF is more advantageous in improving the signal-to-noise ratio of the interferogram and spectra. The results indicate that our system can be applied to atmospheric vertical detection and hyperspectral remote sensing over large area ranges because our denoised technique is suitable for large LWIR FPA.

## 1. Introduction

The interferometric Fourier transform spectrometer (FTS) has greatly improved vertical detection resolution and can accurately obtain the vertical distributions of atmospheric temperature and water vapor profiles because it has more channels, a higher signal-to-noise ratio and a higher spectral resolution than traditional IR hyperspectral sounders [1,2]. Since 2000, in the United States and Europe, infrared (IR) hyperspectral atmospheric sounders (IHAS) have been studied and used for satellite and meteorological observations, such as the IR Atmospheric Sounding Interferometer (IASI) [3], Cross-track IR Sounder (CrIS) [4], Geostationary Imaging Fourier Transform Spectrometer (GIFTS) [5], and Meteosat of Third Generation IR Sounder (MTG-IRS) [6]. In China, the second-generation geostationary meteorological satellite Fengyun-4 (FY-4) was successfully launched in the second half of 2016. The first international Geostationary Interferometric IR Sounder (GIIRS) was carried out for geostationary orbit [7]. It is thought that the developing trend of spaceborne high spectral resolution IR detection technology is following the direction of the large field of view, large dynamic range, high temporal resolution, high spatial resolution, and high sensitivity. Thus, high spectral IR detection performances can be achieved based on integrated large area arrays and miniaturization of the HgCdTe IR detector [8,9,10]. In this paper, we adopt a long wavelength infrared (LWIR) detector with a larger area focal plane array (FPA) based on the GIIRS, independently developed by the Shanghai Institute of Technical Physics, Chinese Academy of Sciences. The LWIR spectral data obtained by the FTS with a large FPA detector can reveal more details about the atmospheric motion information at different altitudes and the distribution of temperature and humidity for the characteristic absorption peak of carbon dioxide. Therefore, it is expected that the data can be used for increasing the accuracy of numerical weather prediction and special weather forecasting [8]. However, it is indicated that the detector noise becomes more significant as the IR FPA scale increases in size. Furthermore, the detectors for FTS require higher response rates than the detectors for IR images. Thus, it is required that the detectors for FTS have a high-speed response and a large dynamic range output to high-frequency signals [11]. As a result, the intensity of the LWIR signals collected by the interference signals acquisition system is weak and mixes with a large amount of noise, which reduces the signal-to-noise ratio of the interferogram. The main sources of the noise consist of the detector, amplifier, AD sampling, and jitter caused by the moving mirror [5,12].

For the purpose of improving the signal-to-noise ratio of interferograms, the interference signals are obtained by using a high-frequency oversampling technique. Due to the high-frequency oversampling, a large number of measured data are collected every time. The data are more numerous than in previous approaches for each pixel of the FPA. In such a case, if the conventional method of data processing is used, it is found that the interferogram composed of primitive measured data shows significant noise after a few experimental measurements. Therefore, an improved method of data processing is required to suppress the noise of the interference signals. Different approaches, such as wavelet transform and empirical mode decomposition methods, are used to eliminate the random noise of interference signals [13,14,15]. However, such approaches are not adept at eliminating noise with frequencies similar to those of the signals. The Wiener [16], Savitzky-Golay (S-G) [17,18], and least-squares smoothing filters [19] are also general and effective methods to remove the incoherent noise from interference signals. In general, the approaches can be divided into spatial and frequency domain denoising. Although the noise can be partly suppressed with the above approaches, the majority of previous studies have not focused on the high-frequency sampling techniques and statistics of noisy measurement data. Thus, it is difficult to find a balance between denoising and overfiltering while using methods such as least-squares smoothing and S-G filtering. To reduce the noise of the interference signals, an accurate model of the denoising algorithm should be built to process the data. Several studies have shown that the KF algorithm is widely used for estimating the measured data of a system when the noise data are known. The KF is a recursion algorithm based on a priori and a posteriori estimation. It is composed of two parts, where the prior estimate of the state is obtained by the prediction part based on the process model, and the posterior estimate of the state is obtained by the update part based on the measurement model. And the accuracy of the algorithm is achieved by controlling the value of the covariance matrix of the measurement and process noises. Zhu et al. [20] proposed an improved KF algorithm to achieve the real-time estimation and correction of noise. Narasimhapp et al. [21] and Gao et al. [22] proposed a weighted covariance adaptive robust KF, which can effectively and accurately estimate the static parameters of the system and control the effects of outliers. The KF algorithm has been widely used in the aerospace, industrial control, communication, economy, and finance industries, among others [23,24]. However, its applicability for reducing the noise of interference signals has not yet been studied. Therefore, we use the Kalman filter (KF) and its improved weighted adaptive Kalman filter (WAKF) to reduce the noise of the multiple measured data for each pixel of the mercury cadmium telluride (MCT) photovoltaic IR FPA detector in our experimental hyperspectral system.

This paper aims to explore high signal-to-noise ratio interferograms for satisfying the development of the future larger area range GIIRS. We emphasize a denoising method for the interference signals of the large LWIR FPA detector. The interference signals are obtained by an FTS consisting of an IR FPA detector and a Michelson interferometer. After the interference signals are rearranged according to specific rules [12], we use the KF to reduce the noise of the multiple measured data of each pixel and then average them to further filter out other noises. To further reduce the effect of overshoot and ringing artifacts on the objective signals, an algorithm based on the KF is used to improve the influence of outliers by adjusting the weighted covariance matrix of the newly constructed sequence. The weight function is constructed based on the principle that the smaller the variance is, the larger the weight. In addition, an adaptive factor is obtained with a three-step method. The effect of anomalous disturbances on the model-predicted numerical values can be controlled by the adaptive factor. We also use the S-G algorithm and the method of averaging the multiple measurements, which are commonly used in the field of processing spectral data, to reduce the noise of the multiple measured data of each pixel. Moreover, the defined reference spectrum is obtained by averaging 100 spectra acquired from the same pixel. The effectiveness of our improved KF algorithm is further illustrated by comparing the interference signals after denoising by different algorithms and by comparing the spectra before and after denoising with the reference spectrum. The results indicate that the KF and WAKF ones can effectively preserve the desired signals and attenuate the random noise significantly. It is found that the WAKF is more capable of improving the signal-to-noise ratio of the interferogram in comparison with the S-G and KF. It is demonstrated that our system can be used for the atmospheric vertical detection and hyperspectral remote sensing of larger area regions for our denoised technique of large LWIR FPA.

The rest of the paper is organized as follows. The experimental procedures are described in Section 2 and Section 3. The results and discussions are presented in Section 4. Conclusions are given in Section 5.

## 2. KF for Interference Signal Noise Attenuation

### 2.1. Experimental Setup and Measurement Solutions

Here, the main experimental setup for acquiring the LWIR interference data is shown in Figure 1. It consists of a Michelson interferometer, an MCT FPA detector, a blackbody, and data processing computers [6]. The MCT materials are grown by the liquid phase epitaxy method, and the mercury composition is about 0.81. The MCT FPA is a planar 32 × 32 array with a center distance of 60 μm. The FPA size is 8630 μm × 9660 μm. Its output data is divided into 8 channels in parallel. In this work, the standard temperature of the detector is ensured by using a refrigerating system. The IR light radiation from the blackbody is separated and then interfered with when passing through the interferometer. The interfered distributions are received by the MCT FPA detector and are converted into analog signals. After being transformed into digital signals with A/D conversion and other circuits, the signals are transmitted to a computer to be saved [11]. The complete interferograms can be obtained after rearranging an initial interferogram file according to specific rules. In this paper, the signal data are collected at sampling frequencies of 80 MHz and 125 MHz when the temperature of the blackbody is 100 degrees. The system is used to demonstrate the applicability of our denoised technique for the atmospheric vertical detection and hyperspectral remote sensing of larger area regions.

### 2.2. State-Space Model for Interference Signal

Theoretically, the collected interfered signals are the sum of the real signals and noise:(1)D=S+N
where S are the real interfered signals, and N is the noise. The purpose of denoising is to recover $\;\hat{S}$ from D as close to S as possible. A state-space model needs to be built for applying KF to the denoising of interference signals. Owing to the high-frequency oversampling technique, the multiple measured data of each collection should change at a rate of zero. The measured error is used as part of the state vector to improve the precision and stability of the algorithm. Consequently, the state-space model for reducing interference signal noise can be expressed as
(2)$$X_{k}=\left[\begin{array}{c} I_{k}\\ b_{k} \end{array}\right]=\left[\begin{array}{cc} 1 & 1\\ 0 & 1 \end{array}\right]\left[\begin{array}{c} I_{k-1}\\ b_{k-1} \end{array}\right]{+w}_{k}$$
(3)$$Z_{k}=\left[\begin{array}{cc} 1 & 0 \end{array}\right]X_{k}{+v}_{k}$$
where $X_{k}$,${\;I}_{k}$,${\;b}_{k}$ are the state, real interference signals and measured error vectors of each collection, respectively, at time k, and $w_{k}$, $Z_{k}$, and $v_{k}$ are the process noise, measurement, and measurement noise vectors, respectively [25]. In this paper, the above state-space model is used for data analysis.

### 2.3. KF for Interference Signal Noise Attenuation

The application of FK for interference signals requires prior knowledge of the dynamic process, the measurement models, and the processing and measurement noises of the interference signals. The linear dynamic system and the measurement model can be written as [26,27]
(4)$$X_{k}=F_{k-1}X_{k-1}+w_{k-1}$$
(5)$$Z_{k}=H_{k}X_{k}+v_{k}$$
where $X_{k}$ and $F_{k-1}$ are the state vector and state transition matrix at time k, respectively.${\;Z}_{k}$ and $H_{k}$ represent the measurement vector and the designed matrix. $w_{k-1}$ and $v_{k}$ are the processing noise and measurement noise vectors, and their covariance matrices are $Q_{k-1}$ and $R_{k}$, respectively. They are independent of each other and have the following characteristics [28]:(6)$$E\left[w_{k-1}\right]=0$$
(7)$$E\left[v_{k}\;\right]=0$$
(8)$$E\left[w_{k-1}w_{k-1}{}^{T}\right]=0$$
(9)$$E\left[v_{k}{\;v}_{k}{\;}^{T}\right]={\;R}_{k}$$
the predicted state vector and covariance matrices of the state prediction errors can be expressed as
(10)$${\;\hat{X}}_{k|k-1}=F_{k-1}{\;\hat{X}}_{k-1}$$
(11)$$P_{k|k-1}=F_{k-1}P_{k-1}F_{k-1}{}^{T}+Q_{k-1}$$
in the above equations, $F_{k-1}$ is the state transition matrix, and ${\;F}_{k-1}{}^{T}$ denotes the transpose of ${\;F}_{k-1}$. ${\;\hat{X}}_{k|k-1}$ is the one-step predicted state vector at time k. $P_{k-1}$ and ${\;Q}_{k-1}$ are the covariance matrices of the state estimation errors and processing noises at time k-1, respectively. $P_{k|k-1}$ is the covariance of the one-step prediction state error based on Equations (4) and (10). Equations (10) and (11) constitute the prediction module of the KF algorithm. Then, the error-correction module of the KF is obtained based on the recursive least-squares algorithm as
(12)$$G_{k}=P_{k|k-1}H_{k}{}^{T}{\left(H_{k}P_{k|k-1}H_{k}{}^{T}+R_{k}\right)}^{-1}$$
(13)$${\;\hat{X}}_{k}={\;\hat{X}}_{k|k-1}+G_{k}\left(Z_{k}-H_{k}{\;\hat{X}}_{k|k-1}\right)$$
(14)$$P_{k}=\left(I-G_{k}H_{k}\right)P_{k|k-1}$$
where $G_{k}$ is the Kalman gain. ${\;\hat{X}}_{k}$ = $\left(\begin{array}{c} {\;\hat{I}}_{k}\\ {\;\hat{b}}_{k} \end{array}\right)$ is the estimated state vector at time k, where ${\;\hat{I}}_{k}$ is the estimated real interference signals and ${\;\hat{b}}_{k}$ is the estimated measured error of each collected data. $R_{k}$ and I are the estimation of measurement noises and identify matrices, respectively. The covariance matrix of the state estimation error is used to calculate the covariance of the prediction state error for the next cycle [22,27,28].

## 3. Wakf for Interference Signal Noise Attenuation

The adaptive Kalman filter is based on residual adaptive estimation (RAE) or innovation adaptive estimation (IAE) [28]. However, with IAE, $R_{k}$ may be a negative definite matrix when the measured data are unstable. Therefore, the WAKF is based on RAE. Then, we consider that the measured data may be anomalously estimated owing to the effects of overshoot and ringing artifacts on the signals. To overcome the anomaly problem, a weighted adaptive method is proposed. The weights are calculated based on the estimated variance of the measurement states. A three-segment approach is used to control outlier perturbations by adjusting the scale parameters [21,27,28]. The error equations for the measurement and predicted state vectors are
(15)$$L_{k}=H_{k}{\;\hat{X}}_{k}-Z_{k}$$
(16)$${\Delta X}_{k}={\;\hat{X}}_{k}-{\;\hat{X}}_{k|k-1}$$

The corresponding covariance matrices can be derived as
(17)$$\sum_{L_{k}}=R_{k}-H_{k}P_{k}H_{k}{}^{T}$$
(18)$$\sum_{{\Delta X}_{k}}=Q_{k}-P_{k}+F_{k-1}P_{k-1}F_{k-1}{}^{T}$$

The optimal estimation for the covariance matrices of newly constructed sequences with the weighted average window method can be expressed as [21,26,29,30]
(19)$$\sum_{L_{k}}=\frac{1}{N}\sum_{j=1}^{N}w\left({\hat{\sigma }}^{2}{}_{k-j}\right)L_{k-j}L_{k-j}{}^{T}$$
(20)$$\sum_{{\Delta X}_{k}}=\frac{1}{N}\sum_{j=1}^{N}w\left({\;\hat{\sigma }}^{2}{}_{{\;\hat{x}}_{k-j}}\right){\Delta X}_{k-j}{\Delta X}_{k-j}{}^{T}$$
where $w\left({\;\hat{\sigma }}^{2}{}_{k-j}\right)$ and $w\left({\;\hat{\sigma }}^{2}{}_{{\;\bar{x}}_{k-j}}\right)$ are the weight functions. ${\;\hat{\Sigma }}^{2}{}_{k-j}$ and ${\;\hat{\sigma }}^{2}{}_{{\;\hat{x}}_{k-j}}$ denote the estimated variance for $Z_{k}$ and ${\;\hat{X}}_{k|k-1}$ at time k − j, respectively, and N is the window width. Based on the principle that the smaller the variance is, the larger the weight, the weight functions are defined as [21]
(21)$$w\left({\;\hat{\sigma }}^{2}{}_{k-j}\right)=\frac{1}{{\;\hat{\sigma }}^{2}{}_{k-j}}/\sum_{j=1}^{N}\left(\frac{1}{{\;\hat{\sigma }}^{2}{}_{k-j}}\right)$$
(22)$$w\left({\;\hat{\sigma }}^{2}{}_{{\;\hat{x}}_{k-j}}\;\right)=\frac{1}{{\;\hat{\sigma }}^{2}{}_{{\;\hat{x}}_{k-j}}\;}/\sum_{j=1}^{N}\left(\frac{1}{{\;\hat{\sigma }}^{2}{}_{{\;\hat{x}}_{k-j}}\;}\right)$$
the sum of the weights is equal to 1, i.e.,
(23)$$\;\sum_{j=1}^{m}w\left({\;\hat{\sigma }}^{2}{}_{k-j}\right)=1$$
(24)$$\;\sum_{j=1}^{m}w\left({\;\hat{\sigma }}^{2}{}_{{\;\hat{x}}_{k-j}}\;\right)=1$$
the approximation formula for ${\;\hat{\sigma }}^{2}{}_{k-j}$ and ${\;\hat{\sigma }}^{2}{}_{{\;\hat{x}}_{k-j}}$ can be expressed as [26,29]
(25)$${\;\hat{\sigma }}^{2}{}_{k-j}=L_{k-j}{}^{T}R_{k-j}L_{k-j}/r_{k}$$
(26)$${\;\hat{\sigma }}^{2}{}_{{\;\hat{x}}_{k-j}}={\Delta X}_{k-j}{}^{T}P_{k|k-j}{\Delta X}_{k-j}/s_{k-j}$$
where $r_{k}$ and $s_{k}$ denote the numbers of measurements and predicted parameters of the state vectors, respectively. $L_{k}$ is the residual vector of $Z_{k}$ expressed by Equation (15), and ${\Delta X}_{k}$ is the residual of $X_{k}$ expressed by Equation (16). Therefore, the newly constructed sequences based on the weighted window function are used to define the covariance matrices $R_{k}$ [28] and $Q_{k}$ of the estimated measurement noise and the processing noise, respectively.
(27)$$R_{k}=\frac{1}{N}\sum_{j=1}^{N+1}w\left({\;\hat{\sigma }}^{2}{}_{k-j}\right)V_{k-j}V_{k-j}{}^{T}+H_{k-1}P_{k-1}H_{k-1}{}^{T}$$
(28)$$Q_{k}=\frac{1}{N}\sum_{j=1}^{N}w\left({\;\hat{\sigma }}^{2}{}_{{\;\hat{x}}_{k-j}}\right){\Delta X}_{k-j}{\Delta X}_{k-j}{}^{T}+P_{k}-F_{k-1}P_{k-1}F_{k-1}{}^{T}$$

In the WAKF algorithm, the adaptive factor of the three-segment function together with the statistic of the predicted state discrepancy is used to avoid determining the parameter values in relying on experience decisions. The adaptive factor $\beta_{k}$ is represented as [26,30,31,32,33].
(29)$$\beta_{k}=\left\{\begin{array}{c} 1,\;\;\left|{\Delta \tilde{x}}_{k}\right|\leq c_{0}\\ \frac{C_{0}}{\left|{\Delta \tilde{x}}_{k}\right|}{\left(\frac{c_{1}-\left|{\Delta \tilde{x}}_{k}\right|}{c_{1}-c_{0}}\right)}^{2},\;\;c_{0}<\left|{\Delta \tilde{x}}_{k}\right|\leq c_{1}\\ 0,\;\;\left|{\Delta \;\tilde{x}}_{k}\right|>c_{1} \end{array}\right.$$
where
(30)$$\left|{\Delta \tilde{x}}_{k}\right|=\frac{\left|\left|{\;\tilde{x}}_{k}-{\;\bar{X}}_{k}\right|\right|}{\sqrt{\mathrm{tr}(\Sigma_{L_{k}})}}=\frac{\left|\left|L_{k}\right|\right|}{\sqrt{\mathrm{tr}(\Sigma_{L_{k}})}}$$
where $c_{0}$ and $c_{1}$ are two criterion constants. Usually, $c_{0}\;$= 1.0–1.5 and $c_{1}$ =3.0–4.5. ${\Delta \tilde{x}}_{k}$ is the learning statistic of the model of predicted state error, and “tr” denotes the trace of a matrix [28,29].

By combining the above weighted adaptive model and the Kalman filter, the improved weighted adaptive Kalman algorithm follows as the prediction equations:(31)$${\;\hat{X}}_{k|k-1}=F_{k-1}{\;\hat{X}}_{k-1\;}$$
(32)$$P_{k|k-1}=F_{k-1}P_{k-1}F_{k-1}{}^{T}+Q_{k-1}$$
and the error-correction equations:(33)$${\;G}_{k}=\frac{1}{\beta_{k}}P_{k|k-1}H_{k}{}^{T}{\left(\frac{1}{\beta_{k}}H_{k}P_{k|k-1}H_{k}{}^{T}+R_{k}\right)}^{-1}$$
(34)$${\;\hat{X}}_{k}={\;\hat{X}}_{k|k-1}+G_{k}\left(Z_{k}-H_{k}{\;\hat{X}}_{k|k-1}\right)$$
(35)$$P_{k\;}=\frac{1}{\beta_{k}}\;\left(I-G_{k}H_{k}\right)P_{k|k-1}$$

In the above algorithm,${\;G}_{k}$ is the modified adaptive Kalman gain, ${\;\hat{X}}_{k}$ is the estimated state vector, and $P_{k}$ is the estimated state error covariance.${\;Q}_{k}\;$and $R_{k}$ are the covariances of the estimated processing noises evaluated using Equation (28) and estimated measurement noises evaluated using Equation (27), respectively. $\beta_{k\;}$ is the adaptive factor evaluated using Equation (29).

## 4. Results

### 4.1. Interference Signal Denoising

An initial interferogram file is generated after the interference signal acquisition system has completed one collection. The new interferogram files are obtained by rearranging the data of the initial file for each pixel. During one motion cycle of the moving mirror, the system collects the data 20,012 times. Based on the high-frequency oversampling technique of frequencies 80 MHz and 125 MHz, 21 and 31 sampling points can be collected each time, respectively. Therefore, the original interferograms with sampling frequencies of 80 MHz and 125 MHz contain 20,012 × 21 and 20,012 × 31 sampling points, respectively.

To verify the feasibility and effectiveness of KF and WAKF in the denoising of IR interference signals, the data of the 747th pixel at the 80 MHz sampling frequency are first analyzed by applying S-G, KF and WAKF. The effect of the S-G algorithm denoising is mainly influenced by the size of the window and the number of smoothing operations. Through several experiments, it is found that the effect of noise reduction is optimal when the parameters of the S-G method are (3, 5), i.e., the width of the smooth polynomial window is equal to 3 and the number of the polynomial is equal to 5. In the KF algorithm, the initial values of noise parameters R and Q are selected as $\left[\begin{array}{cc} 0.01 & 0\\ 0 & 0.01 \end{array}\right]$ and $\left[\begin{array}{cc} 0.1 & 0\\ 0 & 0.1 \end{array}\right]$, respectively, using the trial-and-error method. The S-G and KF are applied to 20,012 groups of sampling points for an interferogram. Then, the next interferogram is obtained by averaging the resulting data. Figure 2 shows the amplitudes of the interferogram as a function of sampling time. It can be found that the noise of the interferogram after KF is more attenuated compared to the S-G algorithm, while some residual noise remains in the signals. A local zooming of Figure 2 is shown in Figure 3 for the convenience of discussion. The result indicates that the interferogram amplitude after the S-G and KF is still not smooth and clear, but the KF is more effective in denoising the interference signals.

In the WAKF algorithm, the covariance matrices of measurement and processing noises are estimated by using the weighted covariance matrix of the newly constructed sequences as Equations (27) and (28), respectively. The weight functions are calculated using Equations (21) and (22), respectively. The estimated value of the signals is updated by using the adaptive factor evaluated with Equation (29). In the algorithm, with the weighted average window method, the window width is varied between 3 and 15. Then, we discuss the effect of different window widths on the performance of the WAKF algorithm. It can be observed that 7 is the optimal choice for statistical smoothing. The values of c0 and c1 are set as 1.5 and 3.5, respectively. Figure 2 presents the noisy interferogram and the denoised interferograms obtained from the S-G, KF and WAKF. The performance of WAKF is compared with that of S-G and KF, as shown in Figure 2 and Figure 3. The result shows that the WAKF algorithm has an obvious advantage over the S-G and KF algorithms in terms of reducing noise and preserving interferogram features in the interferogram at a sampling frequency of 80 MHz.

Since the true value of the signals is not known, the algorithms are evaluated by using the standard deviation (SD). The ideal experimental value should be closely dispersed around the true value. If the SD is larger, the difference between the measured and true values is larger. The SD is calculated before and after the algorithms are applied to an interferogram per
(36)$$\mathrm{SD}=\sqrt{\frac{\sum_{i=1}^{n}{\left(x_{i}-\;\bar{x}\right)}^{2}}{n}}\;$$
where $x_{i}$ is the amplitude of the interferogram, $\;\bar{x}$ is the average value of the interferogram, and n is the length of the interferogram. The results of the denoised signals are tabulated in Table 1. Table 1 illustrates that the SD is smaller after denoising by the WAKF algorithm.

To further demonstrate the interference signal denoising usefulness of our improved KF algorithm, we directly obtain the interferogram of the 747th pixel by averaging the sampling points of each group from the 20,012 groups, as shown in Figure 4. The local zooming of the interferogram is presented in Figure 5 after the denoising of WAKF. It can be seen that the interferogram amplitude of directly averaging the sampling points has abnormal fluctuations near the sampling points 18,000 and the center of the interferogram. The results indicate that the amplitude has significant noise around sampling points 18,000 and 9000 when compared with that of the WAKF algorithms.

Next, we discuss the reliability and accuracy of the denoising methods. The interference signals of the 355th pixel at 125 MHz sampling frequency are denoised by using S-G, KF and WAKF. The real signals are completely concealed in the noise distribution from the whole data, as shown in red in Figure 6. The denoising results after using the S-G, KF and the proposed WAKF are also shown in Figure 6. It can be seen that most noises are reduced in blue and green. Additionally, the results indicate that the denoising of our improved WAKF is better than that of KF. The conclusion is also reflected in the SD result, as shown in Table 2. Here, one can see that the SD of the WAKF is minimal. It can be concluded that our proposed WAKF is more effective than KF in recovering signal characteristics. In combination with the sampling points of the 747th pixel at an 80 MHz frequency, the SDs of interferograms with our WAKF are smaller than those with the KF by 39.49% and 29.22% and smaller than those with the S-G by 69.25% and 46.41%, respectively, for both sampling frequencies.

### 4.2. Denoising Spectra of the 747th Pixel at an 80 MHz Sampling Frequency

The vertical distribution of atmospheric temperature and humidity can be recovered by measuring the radiance of different spectral channels obtained by denoising the collected interferogram data of our LWIR FTS system. Thus, the function of atmospheric vertical detection can be realized [8]. To demonstrate the applicability of our denoising algorithms to atmospheric vertical detection with large FPA detectors, the spectra of the target band are obtained by Fourier transforming the interferograms before and after the denoising algorithm. The reference spectrum is obtained by averaging the 100 lower noisy spectra of the 747th pixel at the 80 MHz sampling frequency. It is known that the large amount of random noise in a spectrum can generally be removed by averaging multiple collected spectra in a certain pixel [10]. As shown in Figure 7, the spectral profile of the target band can almost clearly be seen. Then, the spectra before and after the denoising algorithm are compared with the reference spectrum, as shown in Figure 8. In comparison with the reference spectra, it is obvious that the raw original spectra have considerable discrete noise, and no spectral information for the target band can be obtained at all in Figure 8a,b. In contrast, the noise of the spectra is significantly reduced after using the KF and WAKF algorithms in Figure 8c,d, respectively. The spectral characteristics of the target band can be easily discriminated, as shown in Figure 8d.

The root mean square error (RMSE) is used to further demonstrate the denoising effectiveness of interference signals by our improved KF algorithm, which is defined as follows
(37)$$\mathrm{RMSE}=\sqrt{\frac{1}{n}\sum_{i=1}^{n}{\left(G_{i}-\bar{G_{1}}\right)}^{2}}\;$$
where n represents the number of samples, $G_{i}$ is the spectra value before and after the denoising algorithms, and $\bar{G_{1}}$ is the reference spectrum. The results show that the RSMEs of the spectra with the S-G, KF and WAKF algorithms are reduced by 51.73%, 73.54% and 79.26%, respectively, compared with the original spectrum. And the RSME under the WAKF can further be reduced 57.05% and 21.59% compared with that of the S-G and KF, respectively. The reduction of noise equivalent spectral radiance (NESR) of this pixel [11] occurs due to the reduction of the REMS, which will improve the performance of the LWIR interference signal acquisition system. The results indicate that our denoised technique is suitable for large LWIR FPAs for atmospheric vertical detection and hyperspectral remote sensing over large areas.

## 5. Conclusions

In this work, we explore high signal-to-noise ratio interferograms for satisfying the development of the future larger area range GIIRS. A state-space model is proposed for applying KF to the denoising of interference signals in LWIR FPA interference data. Further. To improve the denoising performance of KF, a WAKF algorithm using weighted adaptive covariance is proposed. In the WAKF algorithm, an adaptive tuning parameter is introduced to adjust for abnormal fluctuations in the measured data. The noise of overshoot and ringing artifacts can be effectively suppressed. Additionally, the WAKF can solve the surplus obvious noise after using KF denoising. In addition, a weight function and an average window method are used to adaptively adjust the covariance matrices of measurement and processing noises. The experimental results of LWIR interference signals with different sampling frequencies and different pixels indicate that our proposed WAKF numerical algorithm is more effective than the S-G and KF algorithms in reducing interference signal noise. Additionally, the SDs of interferograms with both sampling frequencies under the WAKF are smaller than those under the KF by 39.49% and 29.22%, respectively. Moreover, the RSME of the spectra under the WAKF can further be reduced 57.05% and 21.59% compared with that of the S-G and KF at a sampling frequency of 80 MHz. respectively. Furthermore, the spectral characteristics of the target band can be easily discriminated using the WAKF algorithm. The results suggest that the WAKF has better superiority and potential in the attenuation of interference signal noise and the recovery of the ideal interferogram and spectrum. It is indicated that the signal-to-noise ratio of the spectral data obtained by the large LWIR FPA detector can be significantly improved by the denoising algorithms. In conclusion, our constructed system can be used for atmospheric vertical detection and hyperspectral remote sensing over large areas.

## Figures and Tables

**Figure 1 sensors-22-08724-f001:**
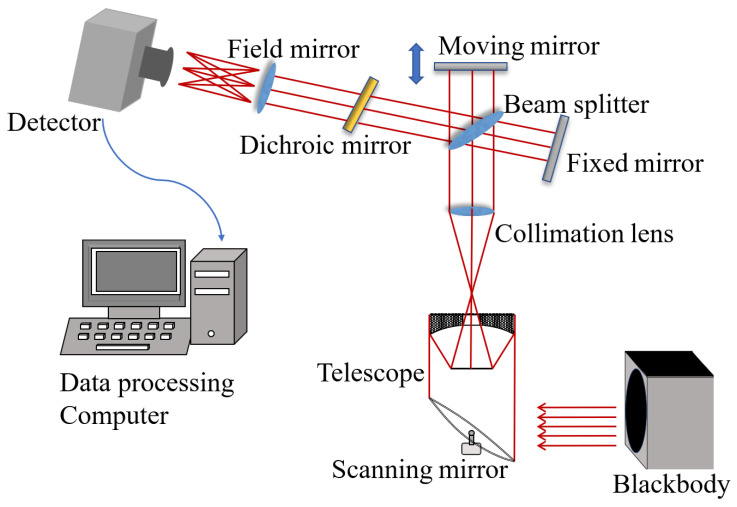
Experimental setup of the LWIR interference signal acquisition system.

**Figure 2 sensors-22-08724-f002:**
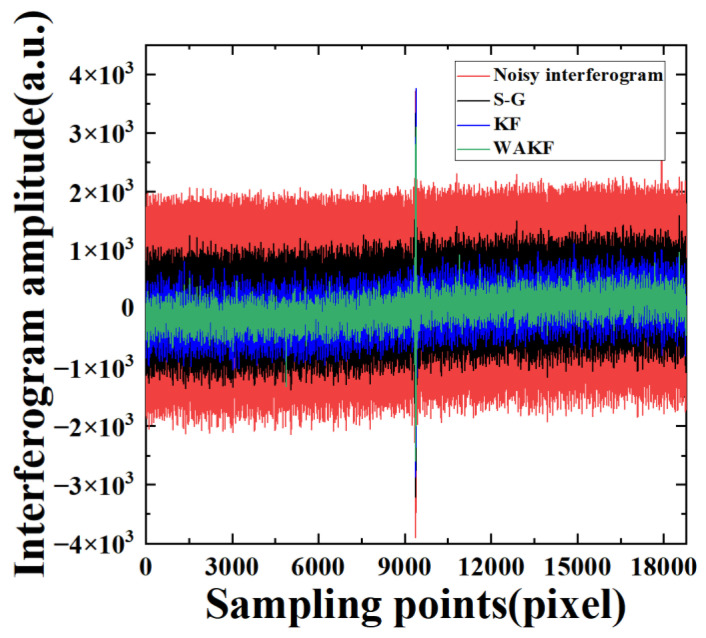
The amplitudes of the interferogram as a function of sampling points for the interferogram of the 747th pixel at a sampling frequency of 80 MHz.

**Figure 3 sensors-22-08724-f003:**
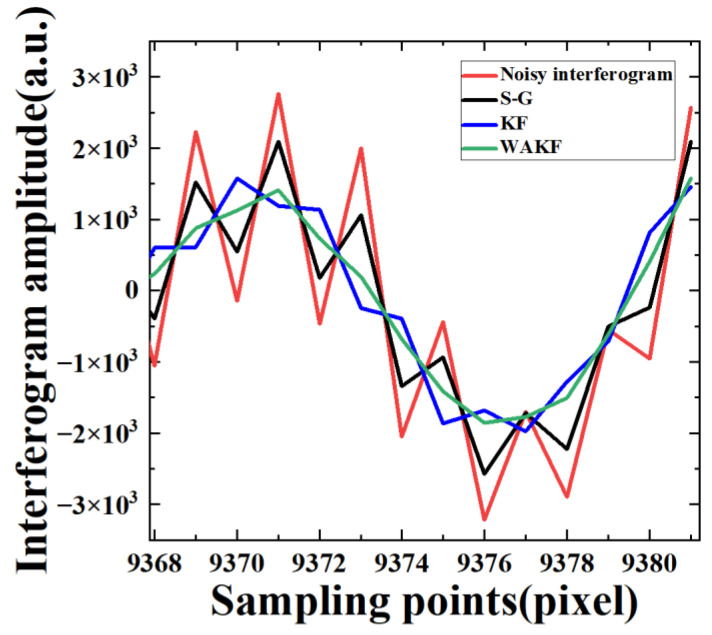
The amplitudes as a function of sampling points for the step interferogram of the 747th pixel at a sampling frequency of 80 MHz.

**Figure 4 sensors-22-08724-f004:**
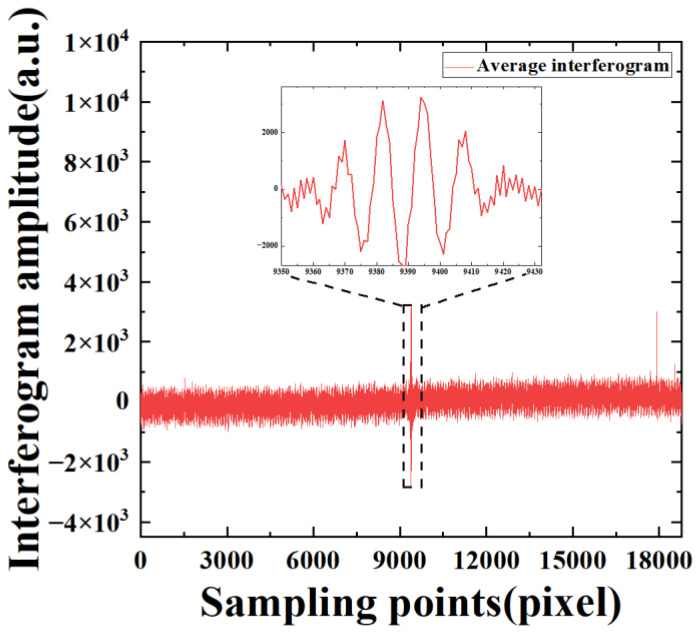
Amplitudes of the interferogram as a function of sampling points for the average interferogram of the 747th pixel at a sampling frequency of 80 MHz.

**Figure 5 sensors-22-08724-f005:**
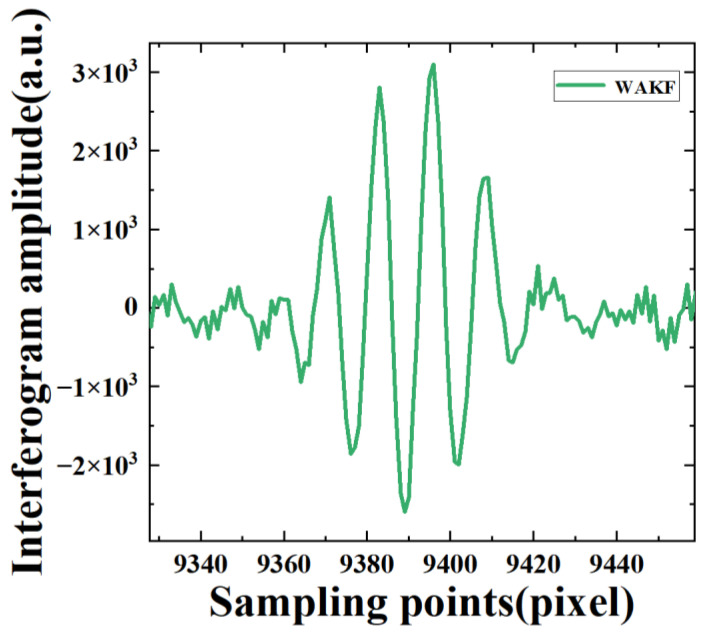
The amplitudes of the step interferogram as a function of sampling points are denoised with WAKF for the 747th pixel at a sampling frequency of 80 MHz.

**Figure 6 sensors-22-08724-f006:**
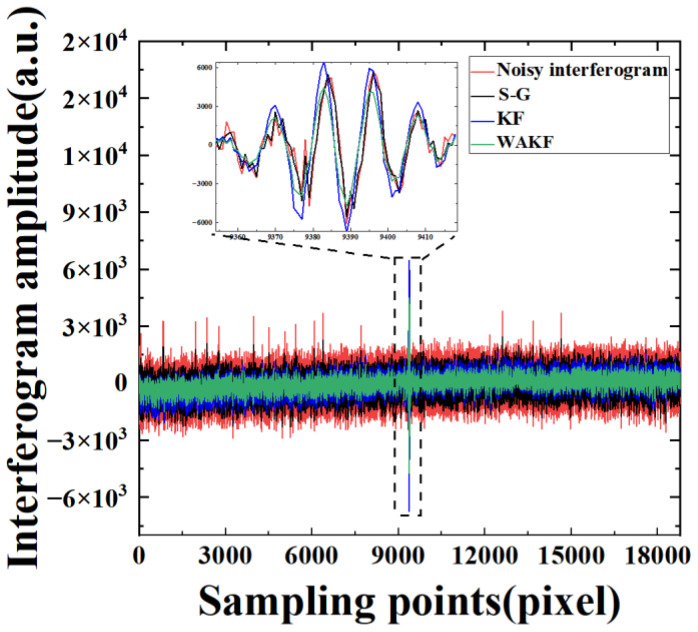
Amplitudes of the interferogram as a function of sampling points for the 355th pixel at a sampling frequency of 125 MHz.

**Figure 7 sensors-22-08724-f007:**
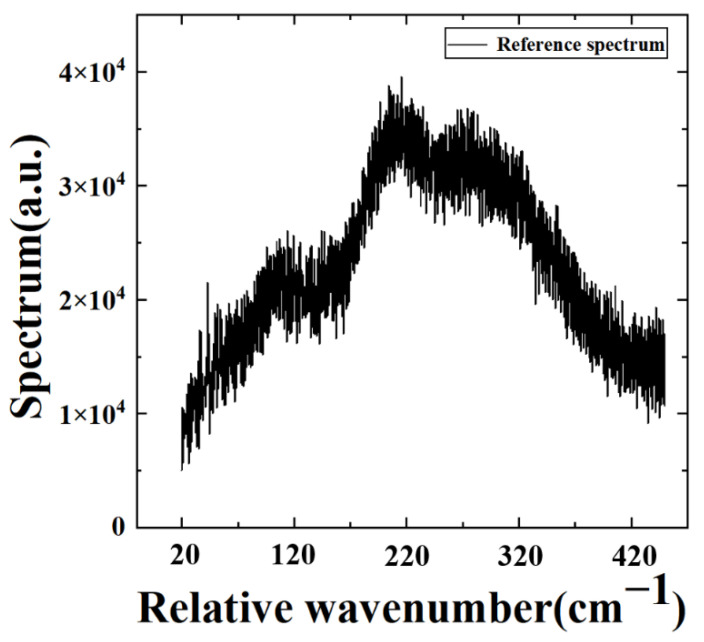
The reference spectrum of the 747th pixel at a sampling frequency of 80 MHz.

**Figure 8 sensors-22-08724-f008:**
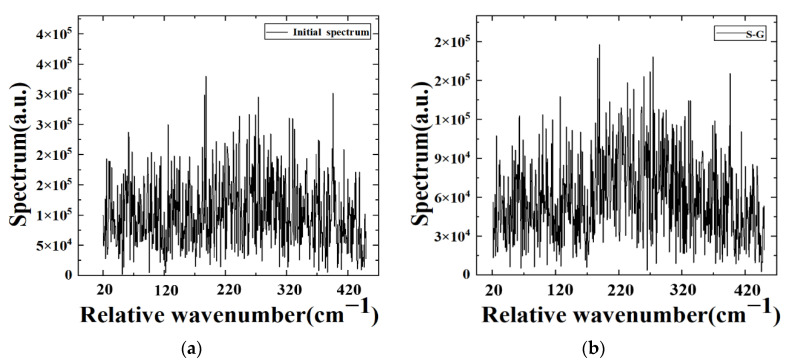
The spectra before and after the denoising algorithms of the 747th pixel at a sampling frequency of 80 MHz. (**a**) Initial spectrum, denoised spectra using the (**b**) S-G, the (**c**) KF and (**d**) WAKF algorithms.

**Table 1 sensors-22-08724-t001:** SD for interferogram of the 747th pixel at a sampling frequency of 80 MHz.

SD	
Noisy Interferogram	1403.30
S-G	733.50
KF	372.83
WAKF	225.58

**Table 2 sensors-22-08724-t002:** SD for interferogram of the 355th pixel at a sampling frequency of 125 MHz.

SD	
Noisy Interferogram	943.62
S-G	673.80
KF	510.12
WAKF	361.10

## Data Availability

Not applicable.

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
