# Peer review of "The Atmospheric Vertical Detection of Large Area Regions Based on Interference Signal Denoising of Weighted Adaptive Kalman Filter"

_sensors, 2022, doi:10.3390/s22228724_

Round 1

Reviewer 1 Report

I would like to recommend this manuscript entitled "The Atmospheric Vertical Detection of Large Area Regions 2 Based on Interference Signal Denoising of Weighted Adaptive 3 Kalman Filter" can be published on Sensors after a revision. My comments are below:

Figure 1 is very poor and hard to understand. I suggest the authors redraw the schematic using professional software, such as 3DS Max, Maya, or PPT. A clear schematic is helpful for readers to understand the paper. The current figure 1 can be put in the Supplementary part or other figures.

Figure 8 is poor. I suggest they redraw this figure and use the same style as other figures.

Reviewer 2 Report

Manuscript ID: sensors-1996603
Type of manuscript:
Article
Title:
The Atmospheric Vertical Detection of Large Area Regions Based on Interference Signal Denoising of Weighted Adaptive Kalman Filter

Authors: Qiying Shen et al.

The authors reported an algorithm based on the improved weighted adaptive Kalman filter (WAKF) to process the long-wavelength infrared spectroscopy measured by a customed Fourier transform spectrometer. The experimental results indicated that this algorithm have the function of excellent noise suppression and improved the signal to noise ratio of the interferogram by 39.50%. Their proposed method is also attractive for spectra data processing. The underlying idea is appealing. However, many figures in the manuscript need to improve their quality to satisfy the requirement of the journal. While the way it is reported in the present manuscript also suffers from some important flaws, which lead me to advise some necessary revisions to the manuscript.

1.      Figure 1 for experimental setup, the figure is hard to read, some background and text inside the picture are all white, which let the reader is hard to discriminate every device of the LWIR interference signal acquisition system. Otherwise, the picture is a little bit mess. Please improve the picture quality by using the better one. While it is better to provide a schematic diagram of the setup, so that the readers will be easier to understand your experimental system.

2.      It is necessary to improve the figure quality of Figs. 2 and 8. The font of title in the vertical axis in Fig.2 is too big, while the resolution of Fig. 8 with 100% zooming is not enough to give a clear appearance. Please revise both figures to better present your results.

3.      Two important figures of Figs.7 and 8, the tick labels of horizontal axis were all missed or did not present. What is the spectra range, how many wavenumbers? Please well organize the spectra in both figures.

4.      It is better to check the grammar and typo errors throughout the manuscript. For example, “Science and_Technology” in line 14 should be written as “Science and Technology”; Line 490,

“Guang pu xue yu Guang pu fen xi= Guang pu” should be “Spectroscopy and Spectral Analysis”.

Reviewer 3 Report

Comments:

Article

The Atmospheric Vertical Detection of Large Area Regions Based on Interference Signal Denoising of Weighted Adaptive Kalman Filter

The article is well-structured and well-written. However, author should add information regarding other filters being used in the introduction part and also justify the reason of selecting and using this specific filter. A brief introduction of working principle should be included in the introduction part.

Some minor comments are:

Grammar and language corrections is need in some parts of the manuscript.

Line 97: Write Narasimhapp et al. [22] and Gao et al. [23] instead of Narasimhapp et al. [22, 23]

Line 409: “In this letter” can be replaced with :in this work”

Reference # 7 has not been cited in the text. In some references, you have used the full form of journal names, but most of the reference contains an abbreviated form. The format should be the same for all references. Some journal names are in italic and some are not. The format should be the same. There should be space between year and volume # in the references. Many mistakes are present in the references. Kindly go through all the references and correct the format.  
